# A Pre-Clinical Study on the Use of the Proprotein Convertase Subtilisin/Kexin Type 9 Inhibitor PEP 2-8 to Mitigate Ischemic Injury in a Rat Marginal Donor Model

**DOI:** 10.3390/ijms26188937

**Published:** 2025-09-13

**Authors:** Maria Antonietta Grignano, Marilena Gregorini, Chiara Barisione, Caterina Ivaldo, Daniela Verzola, Noemi Rumeo, Stefano Malabarba, Maria Chiara Mimmi, Elizabeth Carolina Montatixe Fonseca, Simona Viglio, Paolo Iadarola, Tefik Islami, Eleonora Francesca Pattonieri, Gabriele Ceccarelli, Daniela Picciotto, Giovanni Pratesi, Francesca Viazzi, Emma Diletta Stea, Eloisa Arbustini, Pasquale Esposito, Teresa Rampino

**Affiliations:** 1Unit of Nephrology, Dialysis and Transplantation, Fondazione I.R.C.C.S. Policlinico San Matteo, 27100 Pavia, Italy; ma.grignano@smatteo.pv.it (M.A.G.); t_islami@hotmail.com (T.I.); e.pattonieri@smatteo.pv.it (E.F.P.); e.stea@smatteo.pv.it (E.D.S.); t.rampino@smatteo.pv.it (T.R.); 2Department of Internal Medicine and Therapeutics, University of Pavia, 27100 Pavia, Italy; 3Department of Surgical and Integrated Diagnostic Sciences, University of Genoa, 16132 Genova, Italy; chiara.barisione@unige.it (C.B.); caterina.ivaldo@edu.unige.it (C.I.); giovanni.pratesi@unige.it (G.P.); 4IRCCS Ospedale Policlinico San Martino, 16132 Genova, Italy; 5Department of Internal Medicine and Medical Specialties (DIMI), University of Genoa, 16132 Genova, Italy; daverz@libero.it (D.V.); noemirumeo9@gmail.com (N.R.); francesca.viazzi@unige.it (F.V.); pasquale.esposito@unige.it (P.E.); 6Transplant Unit, Fondazione I.R.C.C.S. Policlinico San Matteo, 27100 Pavia, Italy; s.malabarba@smatteo.pv.it; 7Centre for Inherited Cardiovascular Diseases, Research Department, Fondazione I.R.C.C.S. Policlinico San Matteo, 27100 Pavia, Italy; c.mimmi@smatteo.pv.it (M.C.M.); e.arbustini@smatteo.pv.it (E.A.); 8Cardiac 1 Surgery Unit, Cardiothorax and Vascular Department, Fondazione I.R.C.C.S. Policlinico San Matteo, 27100 Pavia, Italy; elizabeth.montatixe@unimi.it; 9Department of Molecular Medicine, University of Pavia, 27100 Pavia, Italy; simona.viglio@unipv.it; 10Lung Transplantation Unit, Fondazione I.R.C.C.S. Policlinico San Matteo, 27100 Pavia, Italy; 11Department of Biology and Biotechnologies “L. Spallanzani”, University of Pavia, 27100 Pavia, Italy; paolo.iadarola@unipv.it; 12Human Anatomy Unit, Department of Public Health, Experimental and Forensic Medicine, University of Pavia, 27100 Pavia, Italy; gabriele.ceccarelli@unipv.it; 13Unit of Nephrology, Dialysis and Transplantation, IRCCS Policlinico San Martino, 16132 Genova, Italy; daniela.picciotto@hsanmartino.it

**Keywords:** proprotein convertase subtilisin/Kexin type 9 inhibitors, PEP 2-8, kidney transplant, donation after circulatory death, rat marginal donor model

## Abstract

Proprotein Convertase Subtilisin/Kexin type 9 PCSK9 inhibitors (PCSK9i) are a novel class of cholesterol-lowering agents that also offer protection against tissue ischemia by reducing apoptosis, pyroptosis, and myocardial infarct size. This study evaluated the effects of the PCSK9 inhibitor PEP 2-8 during hypothermic perfusion (HP) in a rat model of donation after circulatory death (DCD) kidney transplantation. DCD kidneys were perfused at 4 °C for six hours with either Perf-Gen solution alone (control) or Perf-Gen supplemented with PEP 2-8. Glucose and lactate dehydrogenase (LDH) levels were measured at baseline and after six hours (T6h). At T6h, kidneys were evaluated for ischemic injury, tubular cell proliferation, apoptosis, nitrotyrosine (N-Tyr) staining, tissue ATP and LDH levels, and gene expression of PCSK9 and NOX4. Metabolomic profiling was also performed. PEP 2-8 treatment significantly reduced PCSK9 expression, decreased tubular ischemic injury and necrosis, and lowered LDH release. Treated kidneys showed enhanced tubular cell proliferation, reduced apoptosis, and diminished oxidative stress, indicated by decreased N-Tyr staining and NOX4 expression. Energy metabolism was improved, with higher tissue ATP and glucose levels observed in the PEP 2-8 group. Metabolomic analysis further supported the antioxidant effects of PEP 2-8. This is the first study to demonstrate that PEP 2-8 administered during pre-transplant hypothermic perfusion provides renal protection by improving energy metabolism and reducing oxidative stress in the context of ischemic injury.

## 1. Introduction

Kidney transplantation remains the treatment of choice for patients with end-stage renal disease. However, graft outcomes are often compromised by ischemic injury, particularly in organs from expanded criteria donors (ECD) and donors after circulatory death (DCD) [1]. Worldwide, the use of DCD kidneys has risen sharply, accounting for up to 40% of transplants in some countries as a strategy to address the persistent shortage of donor organs. Despite this growing adoption, DCD kidneys still face high discard rates—ranging from 3% to 33%—primarily due to concerns about ischemic injury and overall graft quality. Although delayed graft function (DGF) is more common in DCD transplants, it is usually manageable and has only a modest impact on long-term survival. Nonetheless, ischemic injury remains a major challenge in transplantation, increasing the risk of DGF, primary non-function (PNF), and acute rejection [2,3,4].

Ischemia-induced hypoxia depletes cellular ATP and elevates intracellular proton and calcium concentrations, leading to mitochondrial dysfunction and activation of programmed cell death pathways such as apoptosis and pyroptosis [5]. Paradoxically, hypoxia also intensifies oxidative stress by increasing reactive oxygen species (ROS) production—particularly superoxide radicals—via oxygen-dependent enzymes including cytochrome c oxidase, NADPH oxidase, and uncoupled endothelial nitric oxide synthase [6]. To mitigate such damage, strategies like machine perfusion (MP) have been developed to limit ischemic injury and assess organ viability prior to transplantation [7,8]. In parallel, pharmacological interventions targeting specific phases of ischemia are under active investigation [9]. In our previous work using a rat model of renal ischemia–reperfusion injury (IRI), we observed a marked upregulation of Proprotein Convertase Subtilisin/Kexin type 9 (PCSK9) following reperfusion, suggesting a potential and previously unrecognized role for PCSK9 in renal IRI [10]. Similar findings have been reported in cardiac [11,12,13,14], hepatic [15] and cerebral [16,17,18] IRI models, where PCSK9 inhibition reduced tissue damage.

PCSK9, a serine protease best known for regulating cholesterol metabolism, promotes degradation of low-density lipoprotein (LDL) receptors, thereby increasing circulating oxidized LDL levels and contributing to dyslipidemia [19,20,21]. While primarily expressed in the liver, PCSK9 is also found in the kidney, intestines, pancreas, lungs, and central nervous system [21,22]. To date, its role has been studied extensively in clinical settings, mainly in the context of metabolic syndrome [23,24]. PCSK9 inhibitors have shown remarkable efficacy in lowering cholesterol, reducing cardiovascular risk [25,26,27,28,29], and attenuating hepatic steatosis [30,31]. Beyond lipid metabolism, emerging evidence links PCSK9 to broader pathophysiological processes [32]. Its expression is upregulated by pro-inflammatory stimuli, [33,34] such as lipopolysaccharide, tumor necrosis factor-alpha, and ROS, and it has been implicated in mitochondrial DNA damage [35,36,37,38,39,40] and pyroptosis, a form of programmed cell death [41,42]. Moreover, studies in fibrosis models have associated PCSK9 with mitochondrial dysfunction, with silencing of the enzyme shown to protect tissue integrity and preserve mitochondrial activity [43,44].

Given the urgent need to minimize ischemic injury in kidney transplantation, this study investigated the therapeutic potential of PCSK9 inhibition during ex vivo hypothermic perfusion (HP) of rat DCD kidneys. We employed PEP 2-8, a synthetic peptide inhibitor of PCSK9 that, in addition to lowering cholesterol, has shown promising efficacy in preventing cardiac IRI and improving endothelial function in other experimental models [45,46,47].

The primary objective was to evaluate whether PEP 2-8 could attenuate renal ischemic injury during HP. A secondary objective was to determine whether PEP 2-8 treatment could enhance graft viability. No predefined humane endpoints were established. If successful, this approach could support the wider use of marginal donor kidneys and substantially improve transplantation outcomes.

## 2. Results

### 2.1. Reduced PCSK9 Gene Expression After Renal Conditioning with PEP 2-8

RT-PCR analysis revealed that kidneys treated with PEP 2-8 exhibited an approximate 30% reduction in PCSK9 mRNA expression compared to paired control (CTRL) (*p* < 0.05, paired *t*-test) (Figure 1).

### 2.2. Reduction in Ischemic Tubular Damage in Kidneys Treated with PCSK9 Inhibitors

Supplementing the Perf-Gen solution with PEP 2-8 conferred significant protection against ischemic tubular damage (median (IQR) CTRL 143.00 (123–165); PEP 2-8 107.00 (82–135); *p* < 0.001) (Figure 2A). Specifically, PEP 2-8 treatment reduced the extent of tubular necrosis (median (IQR): control 12.50% (9.00–15.75%) vs. PEP 2-8 5.50% (3.25–10.50%); *p* < 0.001) while preserving a greater proportion of normal tubular structures compared to controls (median (IQR): control 9.50% (5.50–12.00%) vs. PEP 2-8 29.00% (24.25–39.50%); *p* < 0.001) (Figure 2B–D).

Mild lesions were comparable between groups in terms of severity.

These histological findings were corroborated by biochemical markers of tissue injury; lactate dehydrogenase (LDH) levels were significantly lower in the PEP 2-8 group compared to controls. (Median tissue/effluent LDH levels CTRL 811.30 (range: 378.70–1094) versus PEP 2-8 532.80 (range: 266.20–854.00) (*p* < 0.05) (Appendix A).

### 2.3. Preserved Tubular Regeneration and Reduced Apoptosis in Kidneys Treated with PEP 2-8

To assess tubular regeneration, we measured the expression of Anti-proliferating cell nuclear antigen (PCNA). Kidneys treated with PEP 2-8 showed significantly higher PCNA expression than controls (median (IQR): control 0.61 (0.13–1.65) vs. PEP 2-8 1.96 (0.95–2.58); *p* < 0.05) (Figure 3A). This finding aligns with the greater proportion of preserved tubules observed in the PEP 2-8 group. Furthermore, PEP 2-8 treatment significantly reduced apoptosis. The percentage of kidneys that were positive for cleaved caspase-3 was markedly lower in the PEP 2-8 group compared to controls (control: 56.00% vs. PEP 2-8: 9.00%; *p* < 0.0001) (Figure 3B).

### 2.4. Reduced Oxidative Stress in Kidneys Treated with PEP 2-8

We next evaluated markers of oxidative stress; PEP 2-8 treatment resulted in a significant, approximately 20% downregulation of NADPH oxidase 4 (NOX4) gene expression, compared to controls, key enzyme responsible for generating oxidant molecules ROS implicated in apoptosis and renal damage [48] (*p* < 0.05, CTRL vs. PEP 2-8) (Figure 4A).

Consistent with this, protein tyrosine nitration, a biomarker of oxidative damage was markedly lower in the PEP 2-8 group than controls, as measured by N-Tyrosine immunostaining (median [IQR]: CTRL 7.46% [5.67–9.07%] vs. PEP 2-8 2.85% [2.42–3.31%]; *p* < 0.01) (Figure 4B).

### 2.5. Sustained Metabolic Activity in Kidneys Treated with PEP 2-8

To assess the tissue energetic status, we measured ATP levels in kidney samples at the end of perfusion period. PEP 2-8-treated kidneys exhibited significantly higher ATP concentrations compared to the control group (mean ± SD: PEP 2-8, 0.03 ± 0.01 nmol/μL vs. CTRL, 0.02 ± 0.01 nmol/μL; *p* < 0.05). In parallel, glucose release in to the effluent fluid, reflecting metabolic activity, was significantly elevated in PEP 2-8-group (median [IQR]: PEP 2-8, 35.48% [12.28–82.93%] vs. CTRL, 12.26% [7.10–29.84%]; *p* < 0.01) (Figure 5). In contrast, lactate and pyruvate concentrations in the effluent did not differ significantly between the two study groups.

### 2.6. Metabolomics Profiles of Treated and Control Kidneys

Untargeted metabolomic analysis identified 167 compounds from the reference metabolite library. The resulting relative concentrations were used for statistical analysis. To investigate overall metabolic trends and assess potential group separation or outliers, we performed Principal Component Analysis (PCA) [49]. The analysis revealed no clear clustering between groups, and the first two principal components accounted for only 45.9% of the total variance, indicating a high degree of similarity in the metabolic profiles of PEP 2-8-treated and control kidneys (Appendix A).

To identify specific metabolic changes, the concentration data were analyzed using both unpaired *t*-tests and the non-parametric Kruskal–Wallis test. Across both analyses, only 5 of the 167 detected metabolites showed statistically significant differences between the groups (Table 1 and Figure 6).

## 3. Discussion

In this study, we investigated PCSK9 as a novel therapeutic target to mitigate ischemic injury during ex vivo hypothermic perfusion of kidneys from DCD rats. By targeting PCSK9, we aimed to enhance kidney conditioning and improve organ viability.

We demonstrated that the administration of the PCSK9 inhibitor PEP 2-8 during pre-transplant conditioning phase leads to a significant downregulation of PCSK9 gene expression. This molecular change was associated with tangible benefits, including robust protection of renal tissue from ischemic damage, preservation of tubular proliferation, a reduction in apoptosis, and attenuation of oxidative stress.

In a previous study, we reported that IRI in rats increased PCSK9 expression at both the gene and protein levels [10]. The current study shows that PEP 2-8 treatment can reduce PCSK9 transcription, even after 6 h of cold ischemia. This suggests a novel mechanism by which PEP 2-8 may confer protection against ischemic damage, potentially through direct gene regulation, although off-target effects cannot be ruled out. Our results are consistent with those of Schreckenberg et al. who showed that PCSK9 gene deletion, rather than protein inhibition, significantly reduced cardiac infarct size in an animal model of myocardial infarction [50]. A key finding of our study is that PEP 2-8 treatment significantly decreased the tubular ischemic injury score. Kidneys conditioned with PEP 2-8 retained a higher percentage of normal tubules and exhibited a lower rate of necrosis compared to controls, indicating reduced cellular damage and better preservation of structural integrity. This protective effect extends to programmed cell death, as PCSK9 is known to activate apoptotic signaling pathways during ischemia [23,24,25,28]. In our model, the observed reduction in apoptosis in the treated group appears to be mediated by the caspase-3 signaling pathway. This aligns with in vitro studies, demonstrating that PCSK9 silencing inhibits in oxidized LDL-stimulated HUVECs the caspase-9-caspase-3 signaling pathway, thereby reducing cell damage [51].

Recent studies have also highlighted the role of PCSK9 in promoting oxidative stress, often by inducing NADPH oxidase expression [52,53,54]. Our findings support this association, as PEP 2-8 treatment reduced protein tyrosine nitration and downregulated NOX4—a key member of the NADPH oxidase family implicated in mitochondrial oxidative stress and the pathogenesis of various renal diseases [48,55,56]. This antioxidant effect is consistent with the findings of Liu et al., who demonstrated that PEP 2-8 reduced NOX4 expression and pro-inflammatory cytokine levels in a murine model of atherosclerosis. [57]. The antioxidant properties of PCSK9 inhibition have also been confirmed in other models, including human studies on atherosclerosis-related vascular damage and in rats with post-alcoholic liver fibrosis treated with monoclonal antibodies. [58,59,60,61].

To further investigate the molecular effects of PCSK9 inhibition during organ preservation, we conducted untargeted metabolomic profiling, identifying 167 biochemical compounds. Although stress conditions were the primary driver of metabolic changes, treatment with PEP 2-8 was linked to significant reductions in five metabolites associated with oxidative stress: 2-oxobutyrate, sarcosine, L-carnosine, dimethylamine, and biliverdin.

2-oxobutyrate, an α-keto acid generated by homocysteine metabolism, accumulates when methionine synthase activity is impaired by oxidative stress, reflecting a metabolic shift toward glutathione (GSH) synthesis (Figure 7) [62].

Dimethylamine is a toxic byproduct generated from the degradation of asymmetric dimethylarginine (ADMA) by dimethylarginine dimethylaminohydrolase (DDAH). Under oxidative stress, ADMA levels increase, impairing nitric oxide (NO) production and contributing to endothelial dysfunction (Figure 7) [63,64,65,66]. Supporting this, Moraes Ruberti et al., reported significantly elevated dimethylamine levels in rats following myocardial infarction [67]. Conversely, a reduction in dimethylamine suggests improved vascular homeostasis.

Sarcosine, a metabolite involved in choline metabolism, acts as a weak inhibitor of glycine transporters. Elevated sarcosine has been linked to decreased activity of sarcosine oxidase, an enzyme crucial for protecting cells from oxidative damage. Furthermore, acute sarcosine administration can exacerbate oxidative stress by inhibiting both mitochondrial and cytosolic energy enzymes [68,69].

L-carnosine is a well-known antioxidant, that functions either as a direct ROS scavenger or indirectly by enhancing the Nrf2 signaling pathway [70].

Finally, reduced biliverdin levels indicate decreased heme degradation, consistent with previous reports by Sajid et al., who observed altered heme oxygenase activity under hypoxic conditions [71,72,73].

Together, these metabolite changes reinforce the experimental evidence that PCSK9 inhibition enhances tissue resistance to ischemic injury by reducing oxidative stress.

Beyond its antioxidant effects, PEP 2-8 also appeared to improve energy metabolism in the renal perfusion model. This was demonstrated by increased ATP levels in kidney tissue and elevated glucose release in the effluent, indicating enhanced cellular energy production. Although cellular metabolism is significantly suppressed at 4 °C in PEP 2-8–treated kidneys, it does not completely stop. This residual metabolic activity likely allows processes such as glycogenolysis—the breakdown of stored glycogen into glucose—to continue to some extent.

Furthermore, the reduction in oxidative stress markers suggests decreased cellular damage and more efficient cellular function.

Taken together, our findings indicate that PCSK9 inhibition supports renal energy metabolism by promoting ATP synthesis and limiting oxidative stress.

### Limitations and Strengths of the Study

While this study provides valuable insights, several limitations must be acknowledged. First, the observation period was limited to six hours, which does not reflect the prolonged cold ischemia times typically experienced in human kidney transplantation, often lasting 24 to 36 h. As a result, the experimental conditions may not fully capture the range or severity of injury seen in clinical practice.

Additionally, the study did not examine different doses or administration schedules of PEP 2-8, leaving open the possibility that alternative regimens could yield different or enhanced effects. The investigation was also restricted to the pre-reperfusion phase, so the potential benefits—or limitations—of administering PEP 2-8 during reperfusion or transplantation, when oxidative stress and tissue injury peak, remain unknown.

Finally, the translational relevance of these findings to human kidney transplantation is uncertain. Variations in human physiology, ischemia duration, and clinical factors necessitate caution in directly extrapolating these experimental results to patient care.

Despite these limitations, our study highlights a novel application of PCSK9 inhibitors as a promising therapeutic strategy for various ischemic conditions, with potential for integration into pre-transplant organ conditioning protocols.

Improving the functional preservation of marginal kidneys could significantly expand the donor pool, carrying important implications for transplant programs worldwide. Increasing the availability of transplantable kidneys may help reduce waiting times and decrease reliance on dialysis, ultimately enhancing patient survival and quality of life. Given that kidney transplant recipients generally experience substantially better outcomes than those remaining on dialysis, interventions like PEP 2-8 could play a meaningful role in public health efforts to improve transplantation outcomes.

Moreover, the protective effects of PEP 2-8 may extend beyond kidney transplantation, offering potential benefits in other clinical scenarios marked by ischemia–reperfusion injury.

## 4. Materials and Methods

Figure 8 illustrates the experimental design of this study. The study was approved by the Animal Care and Use Ethics Committee of the University of Pavia, Italy and by the Italian Ministry of Health (protocol code: 175.2023-PR, 3 March 2023). All experimental procedures were conducted in full compliance with the Animal Research. Reporting of In Vivo Experiments (ARRIVE) guidelines [74] ensuring transparency, reproducibility, and adherence to international ethical standards. Animal welfare was prioritized throughout the study, with continuous monitoring and the implementation of humane end-of-life protocols.

All interventions were performed under general anesthesia, causing no significant pain, and were carried out under the supervision of personnel trained in proper animal care and handling. Prior to surgery, animals were housed in approved cages in accordance with current regulations and monitored daily by research staff. Veterinary oversight was maintained throughout the study by the attending veterinarian.

The completed ARRIVE checklist is provided in Appendix A.

### 4.1. Animal DCD Model

Male Fischer rats (*n* = 15) (8 weeks old, Harlan Italy s.r.l. Monza, Italy) were used as a DCD model [75]. Under isoflurane anesthesia (2–5%), a midline laparotomy was performed to expose the retroperitoneal renal areas. The lumbar arteries were isolated and ligated, followed by isolation of the renal arteries and veins. Warm ischemia was induced by clamping the aorta for 30 min to simulate DCD conditions. Subsequently, bilateral nephrectomies were performed, preserving the renal hilum. Following nephrectomy, animals were humanely euthanized under anesthesia using approved methods, in accordance with ARRIVE guidelines [74].

Both kidneys from each DCD rat were perfused for 6 h at 4 °C (cold ischemia) using a continuous falling system (HP) and one kidney from each rat was assigned to a treatment arm:

(A)CTRL (control group, *n* = 15): Left kidneys perfused with Perf-Gen solution (IGL Group, Lissieu, France).(B)PEP 2-8 (treated group, *n* = 15): Right kidneys perfused with Perf-Gen solution supplemented with PCSK9 inhibitor.

HP was performed using a gravity-driven system. Specifically, 100 mL of perfusion solution, with or without PCSK9 inhibitor, was used to perfuse the kidney. The solution was delivered through a 100 mL syringe connected to the organ via a perfusion tube. The syringe was positioned at a height of 2.20 m to achieve a flow rate of 2.5 mL/min and a pressure of 1.81 MPa. ach kidney was placed in a Petri dish resting on an ice block to maintain hypothermia.

A cyclical perfusion system was employed whereby effluent was collected in a beaker beneath the kidney and recycled by refilling the syringe, enabling continuous circulation of the perfusate (Figure 1). PEP 2-8 (10 μM; Sigma-Aldrich, Darmstadt, Germany), a synthetic PCSK9 inhibitor, was prepared by dissolving the stock compound in 1 mL of dimethyl sulfoxide (DMSO) to a concentration of 1 μg/μL. Serial dilutions were performed to obtain a working solution of 2.5 μg/100 μL DMSO. Based on prior IRI studies and the average weight of 8-week-old male Fischer rats (~250 g), a final concentration of 3 μg/100 mL Perf-Gen was chosen for kidney perfusion [45,46,50].

In the control group, an equivalent volume of DMSO was added to the Perf-Gen solution to control for any potential vehicle effects.

### 4.2. Samples Collection

At both the start (T0) and end (T6h) of hypothermic perfusion (HP), 1 mL of effluent was collected using a sterile 1 mL syringe to measure glucose and lactate dehydrogenase (LDH) levels.

At T6h, kidneys from each experimental group were divided into three portions: one was fixed in 10% neutral-buffered formalin for histological analysis, another was embedded in optimal cutting temperature (OCT) compound and frozen for subsequent metabolomic studies, and the remaining tissue was snap-frozen in liquid nitrogen and stored at −80 °C for gene expression and biochemical assays

### 4.3. Tubular Ischemic Damage Score

Twenty subserial cross-sections of formalin-fixed paraffin-embedded (FFPE) renal tissues (*n* = 15 per group) were stained with periodic acid—Schiff (PAS). The tubular ischemic damage (TID) score determined by evaluating all tubules visible in at least 10 non-consecutive, non-overlapping high-power fields as previously described [75].

In tubules examined, epithelial cell flattening (TF), brush border loss (BBL), blebbing (BBF), necrosis (TN), and lumen obstruction (TO) were observed.

TF and BBL were classified as mild lesions and scored as 1, BBF, TN, and TO as severe lesions and were scored as 2. Normal tubules were scored as 0. When ≥2 lesions were present in the same tubule, the highest severity score was assigned.

### 4.4. Tubular Proliferation Index and N-Tyrosine Staining

FFPE sections from 15 control kidneys and from 15 PEP 2-8-treated kidneys were dewaxed in xylene, passed through a graded series of alcohols for clearing, and rehydrated in dis-tilled water. Endogenous peroxidase activity was blocked with 3.7% hydrogen peroxide (H_2_O_2_, *v*/*v* in water) followed by a 15 min rinse in H_2_O. Following three washes in 150 mM PBS, antigen retrieval was performed by microwaving the sections in citrate buffer pH 6 and then incubated overnight at 4 °C with monoclonal mouse anti-proliferating cell nuclear antigen (PCNA) antibody (1:200, Santa Cruz Biotechnology, Santa Cruz, CA, USA) or N-Tyrosine (N-Tyr) antibody (1:100 Santa Cruz Biotechnology, Inc., Dallas, TX, USA). Following three PBS washes, the tissues were treated with the Horseradish Peroxidase (HRP) Goat Anti-Mouse IgG Polymer Detection Kit (IMMPRESS, Vector Laboratories, Inc. Newark, CA, USA) for 45 min at room temperature. The immunocomplex was visualized using a biotin-streptavidin-peroxidase system and 3,3-diaminobenzidine (Dako, Glostrup, Denmark). Sections were lightly counterstained with Harris hematoxylin. Negative controls were prepared by omitting the primary antibody and substituting it with immunoglobulin G (IgG). Ten non-consecutive fields from each immunostained kidney section were analyzed. Images were captured using a Nikon Eclipse E200 microscope equipped with a CCD camera and analyzed with ImageJ software (NIH).

The tubular cell proliferation index (TPI) was calculated as the ratio of nuclei expressing PCNA to the total number of nuclei in each tubule, across all analyzed fields (magnification ×40).

N-tyrosine expression was evaluated by converting the immunohistochemistry images to black and white and quantifying the number of black pixels using ImageJ software (magnification ×10). Results were expressed as the percentage of black pixels relative to the total pixel count [76].

Both TPI and N-tyrosine expression are reported as median and interquartile range (IQR) of the analyzed fields.

### 4.5. Apoptosis

FFPE sections (5 µm) from 15 control and 15 PEP 2-8 kidneys were deparaffinized, rehydrated, and subsequently treated with 3% hydrogen peroxide in methanol. Following microwave antigen retrieval in 0.1 M sodium citrate, slides were incubated overnight at 4 °C in a humidified chamber with a primary polyclonal antibody against Cleaved Caspase-3 (Asp175) (1:400 dilution in PBS containing 0.05% Tween 20) (Cell Signaling Technology, Euroclone, Milan, Italy). After PBS washes, tissues were incubated with UltraPolymer Goat anti-Rabbit IgG (H&L) conjugated to HRP (ready to use; ImmunoReagents Inc., Microtech, Pozzuoli, Italy) for 45 min at room temperature. The immunocomplex was visualized using 3,3-diaminobenzidine (Roche, Microtech). Nuclei were counterstained with Carazzi’s hematoxylin, and apoptosis was examined using a Leica 2500 DM microscope. Negative controls were obtained by omitting the primary antibody and incubating with rabbit IgG (Vector Laboratories, DBA Italia, Segrate, Italy). For each kidney, a total of 1000 tubular cell nuclei were counted, and results were expressed as the percentage of positive cells.

### 4.6. RNA Extraction, Reverse Transcriptase, and Polymerase Chain Reaction

Total RNA was extracted from 15 control tissues and 15 PEP-2-8-treated tissues using the TRiFast II reagent (Euroclone) following the manufacturer’s guidelines and quantified with a BioPhotometer reader (Eppendorf, Hamburg, Germany). RNA quality was also verified assessed by evaluating the 260/280 and 260/230 absorbance ratios, which should fall within the ranges of 1.8–2.1 and 2.0–2.2, respectively. A total of 110 ng of RNA was reverse transcribed into cDNA using the iScript cDNA Synthesis Kit (Bio-Rad Laboratories, Milan, Italy). For quantitative PCR (qPCR), the diluted cDNA was amplified using the LightCycler 96 SW system (Roche, Basel, Switzerland) and Luna Universal qPCR Master Mix (Euroclone) with specific primers. The PCR protocol included an initial denaturation at 95 °C for 120 s, followed by 40 cycles of amplification at 95 °C for 10 s, 59 °C for 30 s, and 68 °C for 20 s. Initially, in accordance with MIQE guidelines, 5 different reference genes were tested: β2-microglobulin (B2M), GAPDH, β-actin, RPLP0, and 18S RNA. Gene expression levels were subsequently calculated as relative quantification using B2M as the reference gene and expressed as fold changes comparing PEP 2-8-treated samples to control kidneys [77,78].
**Gene****Accession Number****Primer Forward 5′→3′****Primer Reverse 5′→3′***Pcsk9*NM_199253CAT GGA ACC TGG AGCGGA TTACC TGG CTA CTT CCGTCA GG*NOX4*NM_053524.1TTT CTC AGG TGT GCA TGT AGCGCG TAG GTA GAA GCT GTA ACC A*B2M*NM_004048.4GGG ACT AAA CCT CCA GCC ACCTA CAG CAC ACG CAGTCT GA 

### 4.7. Biochemical Assays

Glucose and lactate levels in effluent fluids were measured using blood gas analysis (GEM Premier 4000, Werfen, Barcelona, Spain). Glucose release was calculated as the percentage increase from baseline glucose concentrations, based on the difference between glucose levels at the start and end of perfusion.

Effluent pyruvate levels were determined by spectrophotometry (Brea, CA, USA). Tissue and effluent LDH levels were measured using a Clinical Chemistry Analyzer (ARCHITECT, Abbott, Italy). Effluent samples were collected as follows: 15 samples from the control group at T0 and T6h, and 15 samples from the PEP 2-8-treated group at both time points.

Tissue adenosine triphosphate (ATP) levels were quantified using an enzyme-linked immunosorbent assay (ELISA) (ab83355, ATP Assay Kit; Abcam, Cambridge, UK) on homogenized tissue samples (*n* = 15 per group).

### 4.8. Metabolomics Studies

Before analyzing the study samples, the feasibility of performing NMR metabolomics on frozen OCT-embedded biopsies was established by optimizing a protocol for tissue removal from OCT and extraction of hydrophilic metabolites from kidney resections (manuscript in preparation). This optimized protocol was then applied to the 30 study samples (15 from the control group and 15 from the PEP 2-8 group).

To further improve the standard NMR metabolomics workflow, the ASICS approach was implemented for automated metabolite identification and quantification [79,80].

#### 4.8.1. Extraction of Polar Metabolites

To remove OCT, embedded samples were transferred into 14 mL Falcon™ tubes and washed four times with ice-cold PBS, followed by three additional washes with ice-cold H_2_O, keeping the biopsies on ice between each washing step. Samples were then placed to vials containing ceramic beads (CKMix—2 mL, Montigny-le-Bretonneux, France) and homogenized in 1 mL of cold MeOH/H_2_O (80/20, *v*/*v*). The average tissue weight was 171 mg (range 81–257 mg).

Homogenization was performed using a Precellys Evolution Cryolys tissue-lyser (Bertin Technologies, Montigny-le-Bretonneux, France) at 4 °C with three 15 s cycles at 6800 rpm. The homogenate was then centrifuged at 12,000 rpm for 10 min at 4 °C, and the supernatant, containing hydrophilic metabolites, was dried in a Speed Vacuum concentrator (Eppendorf) without heating for 3.5 h.

#### 4.8.2. NMR Spectra Acquisition and Processing

NMR spectra were recorded using a Bruker Avance NEO 700 MHz spectrometer equipped with a TCI CryoProbe (Bruker BioSpin, Karlsruhe, Germany). Data acquisition and processing were performed with the software TOPSPIN 4.1.4 (Bruker BioSpin).

For each sample, one-third of the dried extract was dissolved in 0.2 mL of deuterated phosphate buffer at pH 7.4, containing 50 mM Na_2_HPO_4_/NaH_2_PO_4_, 0.5 mM NaN_3_, and 0.2 mM 3-trimethylsilylpropionic-2,2,3,3-d_4_ acid sodium salt (TSP) as the frequency reference. NMR acquisition began within 30 min of reconstitution, with samples kept on ice. A one-dimensional 1H spectrum was acquired at 25 °C using the 1D-1H NOESY (Nuclear Overhauser Effect Spectroscopy) pulse sequence, along with the standard acquisition and processing parameters optimized for metabolomics of biological samples [81,82].

#### 4.8.3. Metabolite Identification and Quantification Were Performed Using ASICS

The complete ASICS library [79,80] containing 216 reference compounds, was uploaded alongside with the 30 pre-processed spectra of the metabolic extracts. Spectral regions corresponding to TSP, residual OCT, and water (−0.05 to 0.05 ppm, 3.700 to 3.730 ppm, and 4.700 to 5.100 ppm, respectively) peaks were arbitrarily set to 0.00 intensity. Subsequently, spectra were normalized using the Probabilistic Quotient Normalization approach, with the median spectrum as reference [83,84]. Finally, metabolite identification and quantification were performed using the default ASICS settings, applying the advanced joint quantification strategy combined with a first step of independent quantification, with a cleaning threshold set at 1%.

### 4.9. Statistical Analysis

Statistical analyses were conducted using GraphPad Prism software v10 (San Diego, CA, USA). Rats were randomized in a 1:1 ratio to receive Perf-Gen Solution with PEP 2-8 in the left kidney and Perf-Gen Solution in the right kidney, or vice versa, with each rat serving as its own control. The sample size was determined based on the primary endpoint (TID) and on the main comparison between CTRL kidneys and PEP-2-8 kidneys, after 6 h of HP. Due to feasibility, the study included 15 kidneys per group. Assuming 80% power and a two-sided alpha error of 5%, this sample size will allow the detection of an effect size of 1.06 standard deviations, which is considered large according to Cohen’s criteria.

Quantitative data are presented as mean ± standard deviation (SD) or standard error of the mean (SEM), median with interquartile range (IQR), minimum and maximum values, or percentages, depending on the data distribution.

Categorical data are presented as frequencies.

Parametric and non-parametric continuous variables between groups were compared using paired *t*-tests, Wilcoxon tests, or Wilcoxon matched-pairs signed-rank tests, as appropriate. Fisher’s exact test was used to compare frequencies.

Relative metabolite concentrations estimated via ASICS were analyzed using Principal Component Analysis (PCA) after scaling to unit variance [49].

Additionally, univariate analyses, including unpaired *t*-tests and the non-parametric Kruskal–Wallis test, were performed to identify significant differences in metabolite concentrations between PEP 2-8 and control groups.

A *p*-value of less than 0.05 was considered statistically significant.

## 5. Conclusions

PCSK9 inhibitors have previously been shown to enhance organ viability in IRI models involving the heart, liver, and brain [22,33,34,35,36,37,38,39,40,41].

To our knowledge, this is the first study demonstrating that PCSK9 inhibitor PEP 2-8 attenuates ischemic renal injury and exploring its potential use in pre-transplant kidney conditioning. Therefore, PEP 2-8 could be incorporated into mechanical perfusion protocols to enhance the viability and quality of kidneys from donors subjected to prolonged ischemia, especially marginal organs, ultimately improving kidney transplantation outcomes and expanding the donor pool.

## Figures and Tables

**Figure 1 ijms-26-08937-f001:**
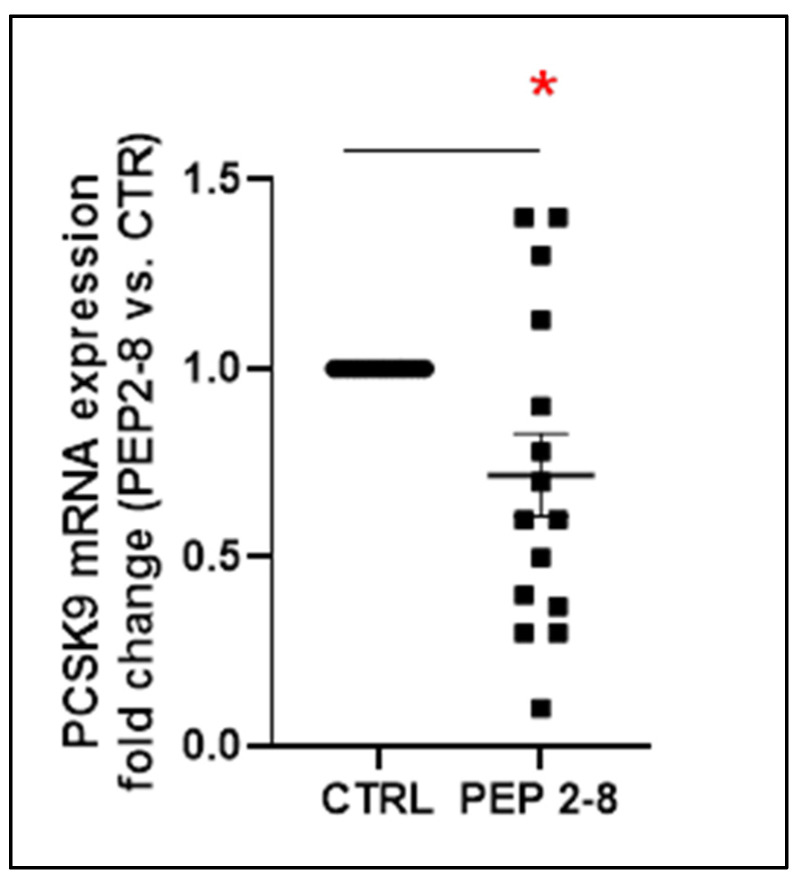
Effect of PEP 2-8 on PCSK9 gene expression. Real-time polymerase chain reaction (RT-PCR) was conducted to assess *PCSK9* gene expression in 15 control kidneys and 15 kidneys perfused under identical conditions, except for the addition of PEP 2-8 to the perfusion solution. All data were analyzed using paired *t*-test. Gene expression was quantified by relative comparison to the reference gene B2M, and results were expressed as fold changes in PEP 2-8-treated samples relative to their paired controls. The results are presented as means ± SEM. (* *p* < 0.05).

**Figure 2 ijms-26-08937-f002:**
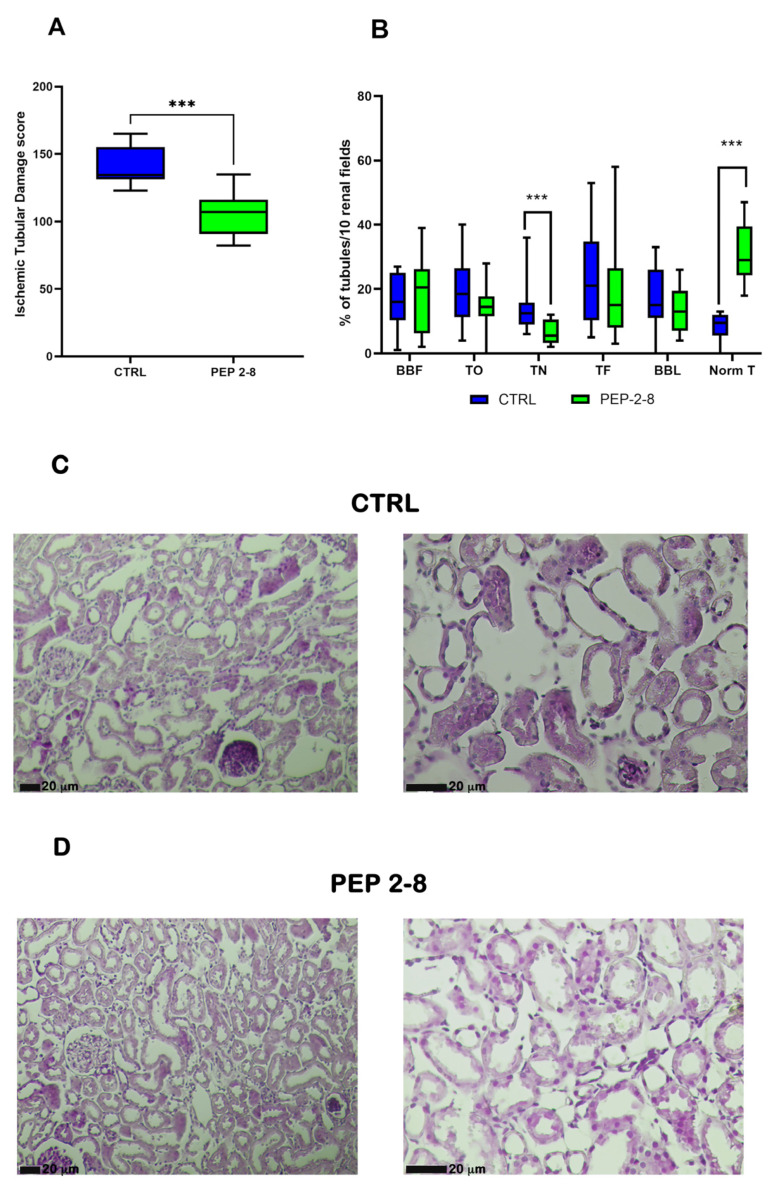
Renal ischemic damage in control kidneys (CTRL, *n* = 15) compared to PEP 2-8 treated kidneys (PEP 2-8, *n* = 15). (**A**) Box-plots representing the tubular ischemic damage score expressed as median and 2.5–97.5 percentile and standard deviation (*** *p* < 0.001). Data were analyzed using the Wilcoxon test. A total of 10 fields per kidney were evaluated. (**B**) The graph represents single tubular ischemic lesions in the study groups. Blebbing Formation (BBF) and Tubular Flattening (TF) were considered mild lesions; Tubular Obstruction (TO), Tubular Necrosis (TN) and Brush Border Loss (BBL) were considered severe lesions; Normal Tubules (Norm T) were scored with 0. Data are expressed as percentage of tubules/10 renal fields and analyzed using Multiple Wilcoxon Test (*** *p* < 0.001). (**C**). PAS staining of representative renal sections from the CTRL group and (**D**) PEP 2-8 group at the end of the perfusion (left: 20×, right: 40× magnification).

**Figure 3 ijms-26-08937-f003:**
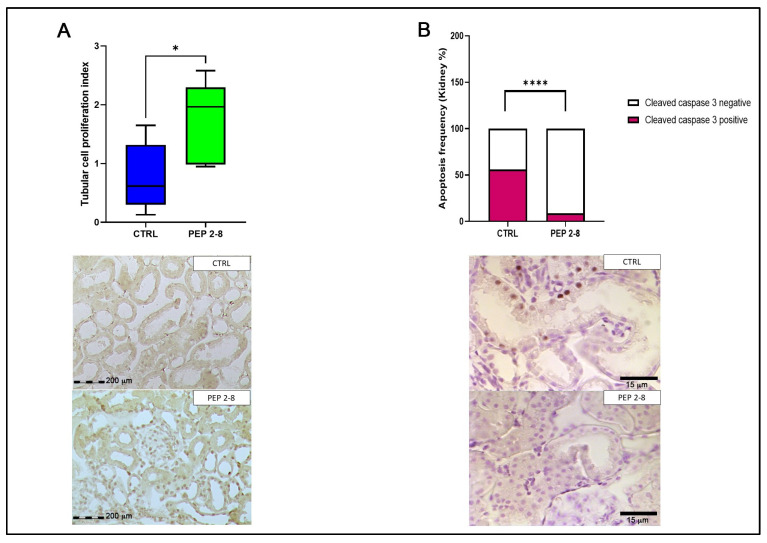
PEP 2-8 effect on cell proliferation and apoptosis in control kidneys (CTRL, *n* = 15) and PEP 2-8 treated kidneys (PEP 2-8, *n* = 15). (**A**) Top: Box plots represent tubular proliferation index (TPI). TPI was defined as the ratio between the nuclei expressing PCNA and the total nuclei in each tubule, in ten non-consecutive fields from each immunostained kidney (×40 magnification). Data are expressed as median and 2.5–97.5 percentile. Wilcoxon test was performed to compare data from the two study groups. (* *p* < 0.05) Bottom: Representative sections of PCNA immunostaining (×40 magnification). (**B**) Top: Apoptosis frequency in the two study groups was assessed by counting a total of 1000 tubular cell nuclei per kidney. Results are presented as the percentage of kidneys with cleaved caspase-3 positive (pink) or negative (white) staining. Data from the two groups were compared using Fisher’s exact test. (**** *p* < 0.0001) Bottom: Representative sections of cleaved caspase 3 immunostaining (40× magnification).

**Figure 4 ijms-26-08937-f004:**
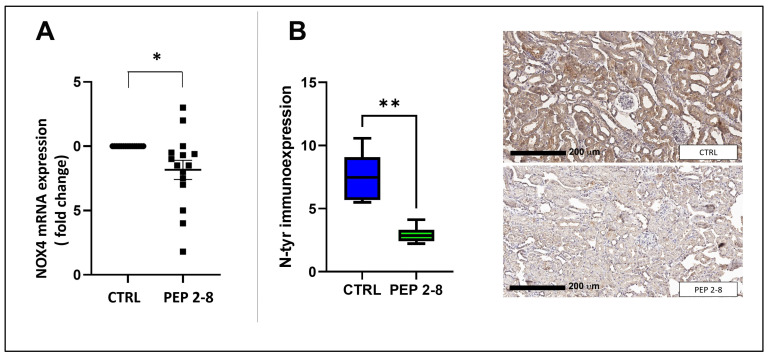
Markers of oxidative stress in CTRL (*n* = 15) and PEP 2-8 (*n* = 15) groups. (**A**) NOX4 mRNA expression was measured by quantitative real-time PCR. Results were normalized to β2-microglobulin (β2-m) transcript levels and expressed as fold changes relative to control kidneys (* *p* < 0.05). (**B**) N-tyrosine expression was quantified by converting immunohistochemistry images to grayscale and analyzing pixel counts using ImageJ software v.1.53a(10× magnification). Data are expressed as the percentage of black pixels relative to the total pixel count. The left panel shows quantitative analysis of N-tyrosine immunostaining, presented as median, 2.5–97.5 percentile range, and standard deviation. Statistical comparisons between groups were performed using a paired *t*-test (** *p* < 0.01). Representative images (20× magnification) are shown on the right.

**Figure 5 ijms-26-08937-f005:**
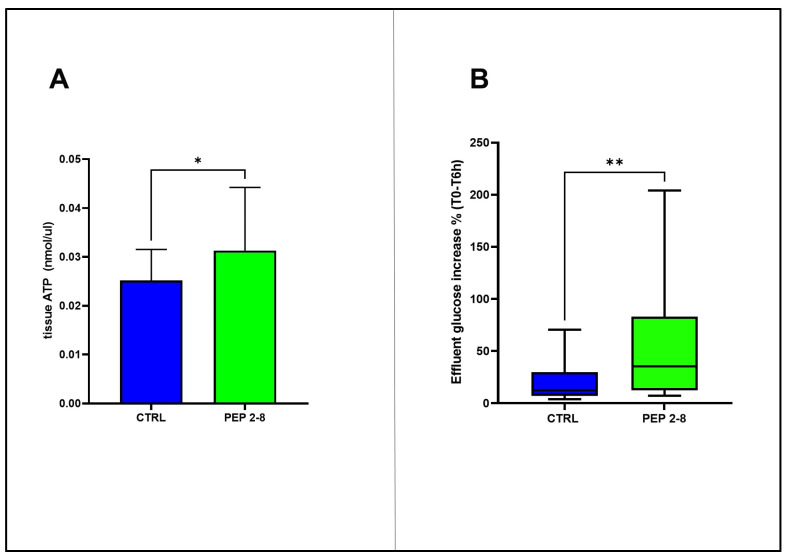
Effects of PEP 2-8 on ATP and glucose levels in the two study groups (CTRL, *n* = 15; PEP 2-8, *n* = 15). (**A**) Columns showing the ATP levels (nmol/uL) in the tissue of CTRL and PEP 2-8 groups, compared suing paired *t*-test. Data are shown as media and standard deviation (* *p* < 0.05). (**B**) Graph represents the percentage of glucose release in the effluent during the hypothermic perfusion. (** *p* < 0.01). Results were analyzed using Wilcoxon test. Data are expressed as median and 2.5–97.5 percentile.

**Figure 6 ijms-26-08937-f006:**
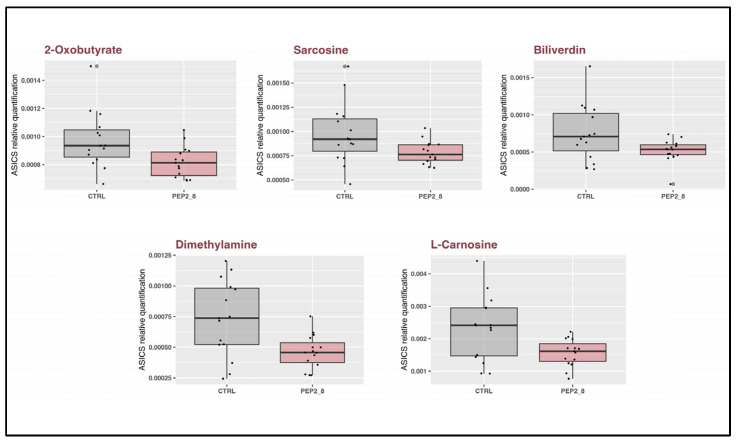
Box plots of the 5 significantly altered metabolites according to *t*-test and Kruskal–Wallis test, relative to 30 samples (15 control and 15 PEP 2-8 samples).

**Figure 7 ijms-26-08937-f007:**
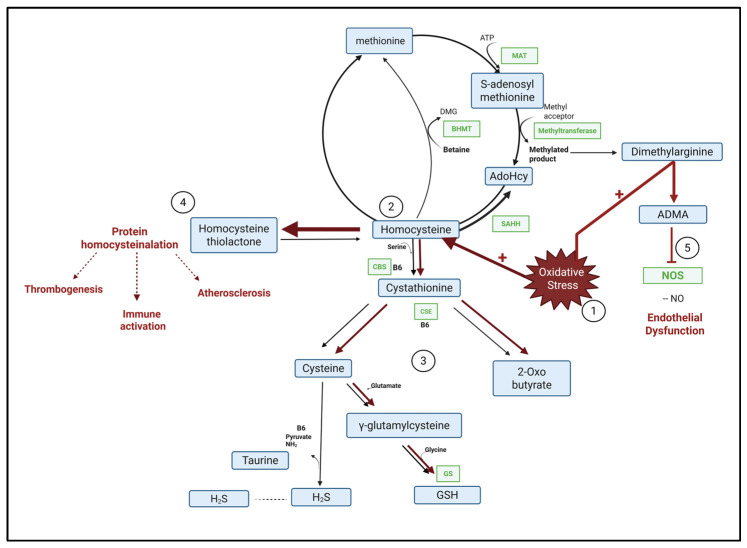
(1) Oxidative stress impairs enzymes like methionine synthase, (2) reducing remethylation of homocysteine to methionine, to produce (3) 2oxo butyrate, cysteine and (4) glutathione, an important antioxidant. Hyperhomocysteinemia accumulates in oxidized forms, (4) promoting oxidative stress leading to endothelial dysfunction, inflammation, and progression of cardiovascular diseases. (5) ADMA, an endogenous inhibitor of nitric oxide synthase (NOS), also increases during oxidative stress, reducing nitric oxide (NO) bioavailability, which impairs vascular function and promotes cardiovascular disease. ADMA: asymmetric dimethylarginine, AdoHcy: S-adenosyl homocysteine, ATP: Adenosine Triphosphate, B6: vitamin B6, BHMT: betaine homocysteine methyltransferase, CBS: cystathionine βsynthase, CSE: cystathionine γ-lyase. GS: glutathione synthase, GSH: reduced glutathione, H2S: hydrogen sulfide, MAT: methionine adenosyltransferase, NO: nitric oxide, NOS: nitric oxide synthase, SAHH: S-adenosyl homocysteine hydrolase. Minus (−) next to substance represents decreased level of substance; Plus (+) represents increased level of substance or increased enzyme activity. Red arrows represent the pathway triggered by oxidative stress. (“->” induction, “−|” inhibition). Created in BioRender. Rampino, T. (2025) https://BioRender.com/xg9tfio (accessed on 10 June 2025).

**Figure 8 ijms-26-08937-f008:**
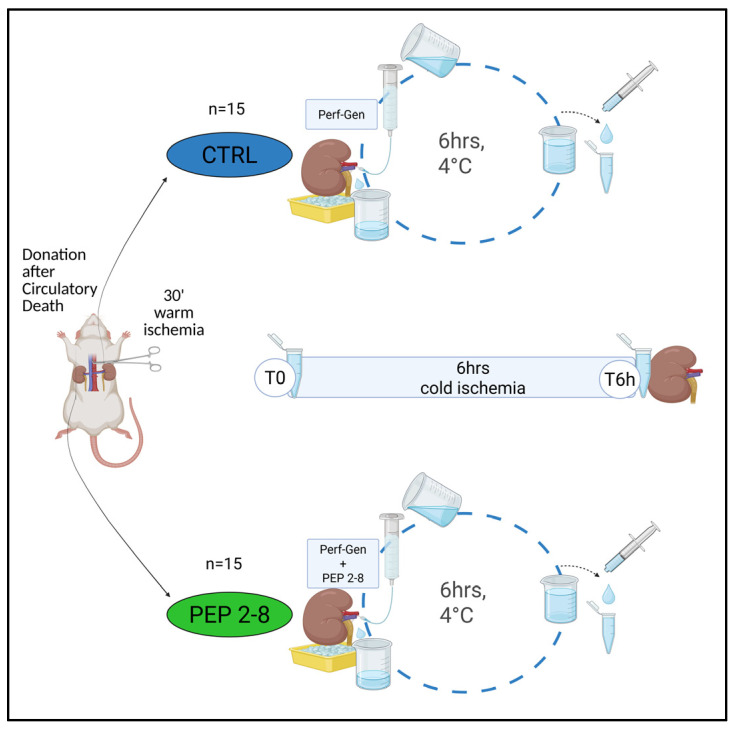
Donation after circulatory death (DCD) was modeled by clamping the rat aorta for 30 min. Following this warm ischemia phase, kidneys were subjected to six hours of continuous hypothermic perfusion at 4 °C using either Perf-Gen solution alone (CTRL group, *n* = 15) or Perf-Gen supplemented with the PCSK9 inhibitor PEP 2-8 (PEP 2-8 group, *n* = 15). Effluent samples were collected at the onset (T0) and conclusion (T6h) of perfusion, while renal tissues were harvested at T6h for further analysis. Figure created with BioRender.com. Rampino, T. (2025). https://BioRender.com/u55x313 (accessed on 10 June 2025).

**Table 1 ijms-26-08937-t001:** Significantly altered metabolites according to univariate statistical tests. The raw *p* values relative to unpaired *t*-test and to Kruskal–Wallis test are reported.

Metabolite	Kruskall Wallis *p* Value	T Test *p* Value
Dimethylamine	0.0136	0.0047
2-Oxobutyrate	0.0152	0.0150
L-Carnosine	0.0213	0.0104
Sarcosine	0.0362	0.0358
Biliverdin	0.0488	0.0322

## Data Availability

Not Applicable.

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
