# Peer review of "A Pre-Clinical Study on the Use of the Proprotein Convertase Subtilisin/Kexin Type 9 Inhibitor PEP 2-8 to Mitigate Ischemic Injury in a Rat Marginal Donor Model"

_ijms, 2025, doi:10.3390/ijms26188937_

Round 1

Reviewer 1 Report

Comments and Suggestions for Authors

This manuscript can contribute to improving the machine perfusion outcome with the PCSK9 inhibitor. The author's suggested strategies and some experimental results are interesting to enhance the quality of the kidney after machine perfusion. This is an interesting study, but I have several questions. I believe that answering the following questions will enable us to have a deeper discussion.

  1. How was the perfusate temperature managed during the supply of a 2.2 m long length tube with a low flow rate ?
  2. How was effect of the concentration of PCSK9 inhibitor?

  1. Can you explain detail process of the glucose release in the effluent during the hypothermic perfusion at 4 °C ?

  1. Is the statistical data processing of CTRL data appropriate in Fig2 and Fig.5A?

  1. In order to reach this conclusion, it may be necessary to conduct detailed comparisons with conventional renal assessment methods in order to verify the results of this study.

Author Response

1: How was the perfusate temperature managed during the supply of a 2.2 m long
length tube with a low flow rate?
We thank the reviewer for this important technical question. Although not explicitly
detailed in the manuscript, the temperature of the perfusate was rigorously maintained
at 4°C throughout the 6-hour perfusion period. Specifically, both the 100 mL syringe
(positioned at 2.2 meters height) and the collection beaker beneath the kidney were
embedded in crushed ice, which was replaced frequently and kept at -80°C before use
to ensure maximum cooling efficiency. Furthermore, the perfusate temperature was
closely monitored every 15–20 minutes using a calibrated thermometer to promptly
detect and correct any deviations. As described in the Methods section, each kidney
was placed in a Petri dish, positioned on top of an ice block. (lines 397-398). During
perfusion to ensure direct and continuous surface cooling. Together, these measures
allowed us to maintain the perfusate at 4°C consistently, minimizing thermal
fluctuations and preserving experimental integrity.
2: How was effect of the concentration of PCSK9 inhibitor?
We thank the reviewer for this insightful question. This study was designed as an
initial proof-of-concept to investigate whether PCSK9 inhibition had a protective
effect in our DCD model, rather than to perform a full dose-response analysis. The
concentration of PEP 2-8 (3 μg/100 mL) was selected based on a comprehensive
review of previous in vivo studies where this peptide demonstrated efficacy in
mitigating ischemia-reperfusion injury in other organs, as cited in our manuscript [42,
43, 46]. While we acknowledge that a dose-response study would provide valuable
additional information, our primary goal here was to establish the therapeutic
potential of this novel approach. We agree that future studies should explore the
effects of different concentrations to optimize the protocol, and we have added this
point to our discussion as a limitation and future direction. (line 329)
3: Can you explain detail process of the glucose release in the effluent during the
hypothermic perfusion at 4 °C?
We thank the reviewer for prompting this important clarification. The release of
glucose into the effluent during hypothermic perfusion at 4°C is primarily driven by
the process of glycogenolysis, which is the breakdown of stored glycogen into
glucose. Although cellular metabolism is significantly slowed at 4°C, it is not
completely arrested. The ischemic stress triggers the enzymatic breakdown of
glycogen reserves within the renal tissue to produce glucose-6-phosphate, which is
then dephosphorylated to free glucose and released. In our study, we interpret the
significantly higher glucose release in the PEP 2-8 treated group (Figure 5B) as an
indicator of a better-preserved cellular metabolic machinery. The treatment likely
protected key enzymes involved in glycogenolysis and glucose transport from
ischemic damage, allowing these cells to maintain a higher metabolic activity and
mobilize energy stores more effectively compared to the control group, where
ischemic injury was more severe. (lines 316-319)
4: Is the statistical data processing of CTRL data appropriate in Fig2 and Fig.5A?
We thank the reviewer for this critical observation regarding our statistical analysis.
In Figures 1 and 4A (ex Fig.2 and 5A), the control (CTRL) data are presented as a
single point at a value of 1.0 on the y-axis because these graphs represent fold change.
By definition, all data from the treated group (PEP 2-8) were normalized to their
paired control kidneys. Therefore, the expression level of the control for each pair is
always 1.0. The graph correctly shows the distribution of the fold changes for the 15
treated samples relative to this normalized baseline. The statistical analysis performed
was a paired t-test, which correctly compares the log-transformed expression values
of each treated kidney against its own paired control, rather than comparing the mean
of the treated group to the value of 1. We believe this approach is appropriate for this
experimental design.
5: In order to reach this conclusion, it may be necessary to conduct detailed
comparisons with conventional renal assessment methods in order to verify the results
of this study.
We fully agree with the reviewer on this crucial point. Our study provides strong preclinical
evidence of the protective effects of PEP 2-8 based on histological, molecular,
and metabolic analyses. However, we acknowledge that these are surrogate endpoints.
The ultimate validation of this approach will require demonstrating a superior
outcome compared to conventional renal assessment methods. In future studies, a
crucial next step will be to transplant these conditioned kidneys and compare longterm
outcomes (such as Delayed Graft Function rates and creatinine clearance) with
organs preserved using the standard of care. This is a key limitation of our current ex
vivo model, and we have explicitly stated this in the discussion section of the revised
manuscript. (lines 331-335)

Reviewer 2 Report

Comments and Suggestions for Authors

Dear Authors,

the manuscript missing in relevant details. I suggest improvements:

Title: missing type of study conducted end not indicated the use of acronims in this section (see keywords too);

Introduction: missing epidemiological data in international and national view, reported in part in the discussion (not suggested). The aims isn't full clear: I suggest classical formula as "the primary aim was... while the secondary was/were...";

Methods: missing international reporting tool for the type of study conducted (e.g. Arrive), mandatary for the international consideration. Attention to clarify the weelness and end of life process for the animals. These elements are fundamental for the ethical consideration;

Discussion; missing clinical practice view of data finding for possible international consideration;

Limits; missing dedicated section;

Conclusion: please according to previous suggestions;

References: are poor in clinical practice view and I reccomand to update over ten years in aren't with high impact of evidence or for Methods support.

Comments on the Quality of English Language

Native English review recommended

Author Response

1. Title: missing type of study conducted end not indicated the use of acronims in this
section (see keywords too).
We thank the reviewer for this valuable suggestion. We agree that the title can be
improved for clarity and adherence to journal standards. We have revised the title to
specify the nature of the study and have removed the acronyms. The new proposed
title is: "A Pre-clinical Study on the Use of the PCSK9 Inhibitor PEP 2-8 to
Mitigate Ischemic Injury in a Rat Marginal Donor Model". We have also updated
the keywords.
2. Introduction: missing epidemiological data in international and national view,
reported in part in the discussion (not suggested). The aims isn't full clear: I suggest
classical formula as "the primary aim was... while the secondary was/were
We thank the reviewer for pointing out these omissions. We have revised the
introduction to include key epidemiological data on the use of marginal donor kidneys
and the clinical impact of ischemic injury, thus better contextualizing the importance
of our study (lines 59-69). Furthermore, we have clarified the study's objectives as
suggested, explicitly stating the primary and secondary aims to improve clarity (lines
107-112).
3. Methods: missing international reporting tool for the type of study conducted (e.g.
Arrive), mandatary for the international consideration. Attention to clarify the weelness
and end of life process for the animals. These elements are fundamental for the ethical
consideration;
We thank the reviewer for this critical and important point. We acknowledge the
omission and have now explicitly stated in the Methods section that our study was
conducted in accordance with the ARRIVE guidelines. We have also expanded the
section on animal care to provide clear details on the measures taken to ensure animal
welfare and the specific procedures used for euthanasia at the end of the experiment,
in line with ethical requirements (lines 357-366 and 382-384).
4. Discussion; missing clinical practice view of data finding for possible international
consideration;
We agree with the reviewer that the clinical perspective of our findings should be
strengthened. We have now added a paragraph to the discussion section where we
elaborate on the potential translational implications of our results. We discuss how
mitigating ischemic injury with PCSK9 inhibitors during machine perfusion could, in
the future, lead to a higher utilization rate of marginal organs and improved long-term
graft outcomes in clinical practice (lines 341-351).
5. Limits; missing dedicated section; Conclusion: please according to previous
suggestions
We thank the reviewer for this suggestion. In the revised manuscript, we have added a
dedicated section titled "Limitations of the Study" to clearly and transparently outline
the constraints of our research, including the use of an animal model and the ex vivo
nature of the experiment (lines 325-335). We have also revised the conclusion section
in accordance with the reviewer’s suggestions (lines 575,576).
6. References: are poor in clinical practice view and I reccomand to update over ten
years in aren't with high impact of evidence or for Methods suppor
We thank the reviewer for this valuable feedback. We have thoroughly reviewed and
updated our reference list to include more recent publications and key clinical studies
that better link our pre-clinical findings to current challenges in clinical
transplantation. This has enriched the context provided in both the Introduction and
Discussion sections. (lines 664-891)

Round 2

Reviewer 2 Report

Comments and Suggestions for Authors

Rest conflicts:

  • please consider native english review in all text, particullary in new part of the manuscript addedd;
  • the reporting tool missing in reference indication in the text and in dedicated section; still missing check list indication in the text and schedule in the supplementary materials;
  • please consider to insert numerical indication for the strenghts and limitations - because you added all two part in the text - and adopt a narrative modalty for description for the limits.
Comments on the Quality of English Language

Native review recommended

Author Response

Thank you for your valuable feedback. We have carefully considered your comments and have made the following revisions to the manuscript:

The entire manuscript, particularly the newly added sections, has been thoroughly reviewed by a native English speaker to improve flow and linguistic accuracy.

We have included the reference indications for the reporting tool both within the text and in the dedicated references section.

The checklist has been added to the main text, and the study schedule is now included in the supplementary materials.

Regarding the strengths and limitations, we have inserted a numerical indication and have adopted a narrative format for describing the limitations, as you suggested.

We are confident that these changes have significantly improved the quality of our work and hope that the manuscript now fully meets your expectations.

Round 3

Reviewer 2 Report

Comments and Suggestions for Authors

Dear Authors,

ready for pubblication. Please adopted classical formula for acronim use of "Animal Research: Reporting of In Vivo Experiments (ARRIVE)" and not "ARRIVE (Animal Research: Reporting of In Vivo Experiments)", line 350.

Best

Author Response

Dear Authors,

ready for pubblication. Please adopted classical formula for acronim use of "Animal Research: Reporting of In Vivo Experiments (ARRIVE)" and not "ARRIVE (Animal Research: Reporting of In Vivo Experiments)", line 350.

Thank you very much for your attentive reading and kind feedback.

As recommended, we have revised line 350 to adopt the classical formula: Animal Research: Reporting of In Vivo Experiments (ARRIVE).

Thank you again for your helpful input.